# Neglected tropical disease meets neglected community: Street children's susceptibility to scabies in Addis Ababa, Ethiopia

**Bewunetu Zewude** [1] *, **Getnet Tadele**[2], **Gail Davey**[3,4]

1 Department of Sociology, College of Social Science and Humanities, Wolaita Sodo University, Sodo town, Ethiopia, 2 Department of Sociology, College of Social Sciences, Addis Ababa University, Addis Ababa City, Ethiopia, 3 Centre for Global Health Research, Brighton & Sussex Medical School, Brighton, United Kingdom, 4 School of Public Health, Addis Ababa University, Addis Ababa, Ethiopia

* bewunetuzewude@gmail.com

**Data Availability Statement:** All the data associated to this study are within the article.

## Abstract

Scabies is a neglected tropical disease (NTD) with high prevalence rate in resource-limited settings. Though street children are susceptible because of lack of sanitation and contact with vectors, few attempts have been made to identify the lived experience of street children with scabies in the global south. This study explored perceived susceptibility to scabies and related lived experiences of street children in Addis Ababa. Using in-depth interviews, we collected qualitative data from selected children of the street to identify their understanding of the causes of scabies, their experiences of managing the condition, and their health-seeking behavior. Informants were recruited to include maximum variation in terms of age, sex, and experience of infestation. The study showed that scabies was common among street children and that the infestation has physical, psychological and social impacts. Study participants believed that scabies had its origin in their living conditions (including poor environmental sanitation and lack of personal hygiene), with lice playing a significant role as vectors of transmission. The informants reported visiting modern healthcare facilities, traditional healers and self-care in response to infestation. By uncovering the embodied experience of a stigmatized skin NTD in a neglected community in the global south, this study contributes to combating neglect and addressing health disparities. Having identified living conditions as the major factor contributing to susceptibility, efforts need to be exerted to change street children's living situations and other structural conditions through reunification with their families or other communities, reintegration and other exit strategies.

## Author summary

Evidence indicates that children living in difficult situations are at high risk of scabies infestation. Street children are one of these groups with higher susceptibility to scabies due to their living conditions. Living as a street child in resource-limited areas substantially increases vulnerability to the condition because of lack of access to healthcare services and other resources to maintain personal hygiene. This study employed a qualitative

Additional data for a research purpose can be accessed without restriction from OSSREA (info@ossrea.net)- an Ethiopian-based research organization up on signing of an agreement to be abided by the relevant policies of publication and other ownership rights.

**Funding:** This research was funded by the NIHR (200140) using UK international development funding from the UK Government to support global health research. The funders had no role in study design, data collection and analysis, decision to publish, or preparation of the manuscript.

approach to understand the susceptibility of street children in Addis Ababa to scabies and their health seeking behavior. The study revealed that scabies is very common among children of the street in the area. The participants associated susceptibility to their living conditions that include sleeping arrangements, sharing of clothes and other materials. They also reported the use of both traditional and modern medicine to treat scabies. The physical, psychological, and social consequences of scabies infestation have also been identified. Although most believe that scabies relates to poor hygiene practices and attempts to prevent the condition have been reported, lack of resources posed significant challenges. We recommend sustainable reunification and reintegration of the street children to help reduce the burden of scabies among street children.

## Background

Scabies is a neglected tropical disease with high prevalence rate particularly in resource-limited areas [1–4]. A range of health complications associated with scabies have been reported, including pain, itching, systemic complications from secondary bacterial infections, nephritis, rheumatic fever, septicemia, glomerulonephritis, and rheumatic heart disease [5–7], abscesses and kidney disease [8]. Scabies also causes a huge public health burden, including stigma, social isolation, absenteeism from duty, and sleep disturbances, and the most serious complications (sepsis, rheumatic heart disease and chronic kidney disease) have fatalities of 5–10% per year [9,10]. El-Moamly [2] noted that scabies may lead to considerable morbidity and mortality in vulnerable individuals, especially in resource-poor communities with less access to healthcare. Engelman et al. [9] demonstrated a substantial socio-economic burden due to absence from employment and education, and the direct costs of accessing healthcare and repeated treatments. As morbidity partly depends on access to quality healthcare, people in resource-limited settings such as Ethiopia are more susceptible [11].

Scabies is one of the most common public health concerns in Ethiopia with an overall prevalence of 14.5% [8]. According to Enbiale and Ayalew [12], scabies prevalence was 45.9%, 48.1%, and 18.6% among children under 2 years, from 2 to 18 years, and over 18 years, respectively, in a large outbreak in Amhara region. A number of community-based studies have shown that factors such as low income, using only water for hand washing, using unimproved water sources, not washing, changing clothes infrequently, contact history with confirmed scabies patients, sharing beds, sleeping on the floor, living with internally displaced people, having a large family size, and the presence of animals at home are predictors of scabies morbidity [8,10,13–15]. Overcrowding is believed to have a significant effect on the spread of scabies, reflecting the fundamental role of person-to-person or skin-to-skin transmission [16,17].

According to WHO [4], scabies and other neglected tropical diseases flourish in settings where access to clean water, sanitation, and quality healthcare is scarce. Social groups at risk include people whose routines and living conditions are linked to poor personal and environmental hygiene, such as young children, the elderly in nursing homes, prison inmates, adolescents, and people living in refugee camps [5–7,16,18]. Children in resource-poor areas are at higher risk [4]. Chowdhury et al. [19] found that street children in Bangladesh are exposed to various skin diseases and other communicable infections due to their overcrowded living situations, unhealthy sleeping areas, irregular baths, and fewer changes of clothes.

Other studies [e.g., 20–23] have shown that the majority of street children suffer from skin problems including scabies, pyoderma, dry skin, furunculosis, and dermatitis. A study by Malgaonkar and Kartikeyan [24] showed significantly higher frequencies of scabies among

children living on the streets in Mumbai compared to those in NGO-run open houses. A recent study in Ethiopia by Zewude et al. [25] revealed that street children are highly susceptible to skin diseases and other vector-borne infections due to crowded sleeping arrangements and the absence of the means to maintain personal hygiene.

With the exception of a few studies, the topic of street children's vulnerability to scabies and associated health-seeking behavior has received limited attention from health professionals and researchers [5,10]. The few existing studies are dominated by medical perspectives, and studies undertaken from social science perspectives are lacking. Although the prevalence of scabies among certain groups of children in Ethiopia has been researched [e.g. 10,13,14,15,26], studies conducted on children living on the street and under other difficult conditions are scarce in Ethiopia. Since street children face a heightened risk of scabies because of their living conditions, it is crucial to study and gain insight into their embodied experience of scabies. This paper explores the susceptibility to scabies and related health-seeking behavior of street children in Addis Ababa, Ethiopia in a bid to inform viable interventions and help reduce neglect and health disparity.

## Materials and methods

### Ethics statement

The nature of the study itself, studying children of the street without adult supervision made it difficult to obtain informed consent from their parents or legal guardians. Contacting the [biological] parents of the children for consent was not possible as most of them came from remote rural areas. Hence, we obtained the verbal consent of their guardians on the street. In addition, we utilized other approaches to ensure the safety, privacy, and confidentiality of the study participants. Potential participants were provided with information about the research, the level of potential risks involved as a result of participating in the research, the type of information required from them, and efforts used to maintain confidentiality. Our data collection protocol included an informed assent form containing the information that the children required, including that they had the right to decide to participate or not, that the interviewer would still take good care of them no matter what they decided, and that they could let the interviewer know if they would like to take a break or stop at any time. The fundamental principles of "do no harm" and "the best interest of the child" were seriously maintained throughout data collection. All interviews were conducted individually in a private location, enabling respondents to feel that they were in a safe environment. We maintained confidentiality by de-identifying respondents in this report. Further, the main project entitled: Social Sciences for Severe Stigmatizing Skin Conditions (5S Foundation) on which this study is a sub-set, obtained ethical clearance from the Research Governance and Ethics Committee (RGEC) of Brighton and Sussex Medical School (BSMS) (Reference: ER/BSMS9E3G/8).

### Study area

The study was conducted in selected areas of Addis Ababa city. Being the capital city of Ethiopia, Addis Ababa carries the greatest share of the urban destitute in the country [27]. The number of street children in Ethiopia and particularly in Addis Ababa is increasing as prevailing socio-economic situations put many children at risk of joining street life [28,29].

Although the mobility of the study population results in changing patterns of distribution across the city, there are some areas in the city identified as hot spots. From the 11 sub-cities of Addis Ababa, street children are more concentrated in *Addis Ketema*, *Yekka*, *Bole*, and *Kirkos* sub-cities, according to a scoping study of the Ministry of Labour and Social Affairs [27]. From these, we randomly selected *Bole* and *Kirkos* sub-cities mainly based on our previous

fieldwork experience in these areas [e.g. 25]. Other related studies [e.g. 30,31] also confirm that the target groups are more concentrated in these areas.

## Selection of study participants

We entertained as much flexibility as possible in setting inclusion and exclusion criteria. The major inclusion/exclusion criteria were 1) being a child of the street, and 2) being below 18 years old at the time of data collection. Since the study primarily involved collecting data about the embodied experience of scabies infestation, street children who reported a history of a previous scabies infestation or those infested with it during the interview were purposely selected using a snowball sampling technique. In addition, street children with no history of infestation were also included. There is a consensus in the literature, including Ethiopian legal documents, that 18 years is the legal age of adulthood. We included street children who could express themselves and were below the age of 18 years. Accordingly, children between 11 and 17 years participated in the study.

The other important inclusion/exclusion criterion was the classification of the study population into "children of the street" and "children on the street" depending on their level of involvement in street life. The former are children who have left their families and for whom the street serves both as a source of shelter and livelihood (complete involvement) [32]. This excludes children who work or spend a significant proportion of their time on the street but return home for the night, have frequent family contact and close adult supervision and are known as "children on the street". This study included only children of the street because of their comparative disadvantage in terms of access to basic resources to maintain personal hygiene and their sleeping arrangements. To obtain diverse perspectives and lived experiences, we considered maximum variation in research participants in terms of age, sex, residential background, and number of years lived on the street.

## Method of data collection and analysis

Qualitative data were collected using in-depth interviews. We prepared an interview guide containing unstructured questions. GT and BZ jointly prepared the data collection instrument, while BZ collected the data under the close supervision of GT. The protocol was organized into three major sections (introductory, the main section, and closing) that include other subsections. The introductory section dealt with opening questions prepared with the dual purpose of building rapport and collecting data about their socio-demographic characteristics including area of origin, issues that forced them to move to the street, and the major health problems they faced. The major opening questions were "Would you please tell me a few things about yourself, such as your name, area of origin, etc?" and "What are the common health threats you encounter while living on the street?". The second section was designed to answer the specific research questions and was further classified into 1) participants' perceived susceptibility to and lived experiences with scabies, 2) perceptions about risk factors associated with the condition, and 3) their health seeking behavior in relation to scabies. Some of the questions in this section include: "How prevalent is scabies among children in your situations (living on the streets)?", "Do you believe that you are susceptible to scabies and why?", "Is there any association between scabies and living/working situations on the street?", "What do you think causes scabies?", "Do you think scabies can be treated and how?", "What should a street child do when infested with scabies?", "If ever infested with scabies, what did you do?", and "How are children infested with scabies treated by other children that are not infested with it?"

The interview was undertaken in private locations suggested by the study participants. Before beginning the interviews, we tried to build rapport with the interviewees and the interviews were conducted in the form of informal conversations. Because many of the children who had been on the street for less than a year came from Oromia region and spoke only their mother tongue language (Afaan Oromoo), we interviewed them in their own language, which also contributed to building rapport. Although most street children had diverse ethnic identities, if they had been on the street for many years, they were well integrated into the dominant sub-culture of the street in Addis Ababa, and spoke the Amharic language. Therefore, Amharic was used as the language for most interviews.

Because the period of interviews (December, 3–28, 2023) was when the government was preparing for the African Union annual summit in Addis Ababa, the street children were trying to avoid capture by the police so the timing was not ideal for the data collection. The police regularly rounded up and either relocated or temporarily held street-based children, beggars and sex workers who work alongside the major roads of the city prior to international meetings such as those of the African Union or other high level international meetings and conferences that are frequently hosted in Addis Ababa. While it is not made clear as to why these round-ups occur, it is often presumed that this is done either out of concern about security, "the image of the city", or to ensure that street-based children, beggars and sex workers do not hassle delegates and participants during these occasions. The children's anxieties about the possibility of arrest made it difficult for us to choose appropriate locations for the interviews -cafés and small pubs were not sufficiently discreet. Most children preferred to be interviewed in parks or underground places (such as under highway bridges) where they slept at night. Most of the children were addicted to glue which made them semi-conscious and difficult to interview. We found that approaching them before they collected enough money to buy glue (mostly before10:00a.m) was the best time to interview. After 5:30 pm was also convenient because most would invest the money collected then on food and bedroom costs rather than glue.

The number of children interviewed was determined based on the principle of theoretical data saturation. Data were considered saturated when no new elements were found and the addition of new information was no longer necessary, since it would not alter the comprehension of the researched phenomenon [33]. We purposefully selected the first interviewee and continued to interview up to 25 children at which point, we generated no new information. The average duration of the interview was 28 minutes—most participants got bored after some minutes of conversation. All interviews were audio recorded; BZ carried out the transcription of the recordings and all subsequent activities with guidance and supervision by GT. We conducted deep readings into the transcripts and field notes followed by coding using NVivo 12 software. We employed a mixture of inductive and deductive approaches to coding. The data collector started setting codes deductively based on the interview guide and field notes. This was followed by inductively developing new codes and iterating on the existing ones as the transcripts were sifted through. In a bid to reduce the challenge of subjectivity and provide a sound interpretation of data, the data analysis involved collaborative coding. Although much of the iterative process of coding and interpretation of the data ended with an acceptable agreement among the coders, some of the inter-coder disagreements were resolved through discussion.

The coding generated three major themes: susceptibility to scabies and related lived experiences of infestation with scabies, understandings about the epidemiology of scabies, including its causes and symptoms, and primary and secondary health-seeking behavior. Additional sub-themes under each of these themes were identified and interpreted. Data are presented in the form of narrations along with selected quotes.

## Results

The study participants had diverse socio-demographic characteristics. Most (63.6%) of them came from different areas of Oromia region including Ziway (Batu), Shashemene, Jimma, Assela, and Hararghe. The second largest number of participants came from Sidama region, followed by those from Addis Ababa and Amhara region (Debrebirhan). Street girls constituted less than a quarter of the total participants (21.7%). Because females were not visible on the streets, we had to make additional efforts such as going to their sleeping areas (DC- a slum area in *Kirkos* sub-city having one of the cheapest bedrooms (0.89$/night) with multi-layered beds) to find them. The highest level of education of most (73.9%) participants was primary, and only two reported to have been enrolled in grade 9. Five had never attended formal education. The age of participants was between 11 and 17 years and the length of time they stayed on the street ranged from 2.5 months to 11 years.

Here, we present the major findings pertaining to study participants' lived experiences of infestation with scabies, their understandings of the epidemiology of the condition, and their health-seeking behavior.

### Susceptibility to scabies and lived experiences of infestation

The study indicated that street children in Addis Ababa are susceptible to scabies. Not only did many study participants report previous experience of infestation with scabies, we also observed symptoms of both active and recovered infections during the interviews. The interviewees explained that scabies was a common health problem among street children in the study area:

> *I see many street children infested with scabies. Not only are many children infected with it, it is also evident that those who have been infected experience repeated infestations. For instance, I was infected 3 times [only] after joining street life. (XX3, 14, M)*

Scabies is widely known among the street children by its nickname "*Maal naawayyaa*" a phrase taken from Afaan Oromo meaning "what shall I do!?" According to the informants, this name was given because of the symptoms of the disease, particularly intense itching, and was an Afaan Oromo term because most children living 'of the street' in Addis Ababa are from the Oromo ethnic group. Nevertheless, we found that this nickname was shared by children with non-Oromo ethnic identities. We also found children using the Amharic term "እከክ" ("*ekek*") and the common Oromo term "*Cittoo*", both referring to scabies.

There was a widely held perception that scabies increases during the dry season. Many associated the dry season with the presence of dust. In addition, the dry season was linked with higher temperature, thought to create favorable conditions for the breeding of the agents causing scabies:

> *Scabies increases especially during the dry/warm season. During the rainy season, we are more susceptible to malaria and other health problems than the scabies. (XX11, M, 15)*

> *The hot weather of the dry season is convenient for the lice to breed and spread. (XX2, 11, M)*

Participants thought that susceptibility to scabies varied in different settings. Dirty physical environments such as contaminated rivers or ponds, and crowded areas have been identified as areas where scabies become more common:

*Bedrooms are high risk areas for the transmission of scabies. There is no louse when you sleep outside because we don't always sleep in the same place. There are many vectors, such as louse in the bedrooms. If you have slept there even with your neat clothes, it is inevitable that you will be contaminated. (XX4, 17, F)*

*Scabies is more prevalent around polluted river areas. For example, if you wash your body in dirty river water, you will soon be infected with scabies. In addition, scabies also exists around other dirty areas (XX5, 14, M)*

Both physical and emotional symptoms of scabies were reported. Participants said that scabies infestation could be physically identified by the appearance of itchy rashes around the palms and ankles, followed by burrows after intense itching. In addition, painful experiences were reported, especially when fluid entered into the cracks:

*At the beginning, a small scar-like swelling appears on your skin and you start to itch it. XX11, M, 15)*

*It burns you as if you are in fire and it pierces you like a needle. (XX6, 14, M)*

*It burns a lot when fluid such as water enters into the cracks of your hand. (XX3, 14, M)*

*After scratching it, severe pain follows (XX9, F, 16)*

The most commonly reported symptom was intense itching followed by bleeding and a small sore appearing around the palm and between fingers. Research participants disclosed that the intensity of itching increased at night with significant impact on the quality of sleep:

*It simply urges me to itch it again and again. I couldn't sleep at night as the severity of its itchiness aggravates at night. (XX4, 17, F)*

*Scabies involves involuntarily scratching your body followed by bleeding of the part you have been scratching. (XX5, 14, M)*

Participants also believe that it takes some time for scabies to cause symptoms after a person is infested:

*The fact that the fluid from the infected person has contaminated your body doesn't mean that you will immediately start scratching; it takes some time. (XX5, 14, M)*

The participants seem to have experienced the physical, social, and psychological impacts of scabies infestation. It was reported that in addition to pain, illness episodes cause isolation and limit one's capability to be involved in the usual social and personal activities, including eating with other children:

*They (other street children) labeled me as a patient and they have been bringing food separately for me. I had no chance to eat with the other children as usual because of the scabies. (XX3, 12, M)*

*You can't even eat food as you wish: when the stew or water touches the crack-like burrows between your fingers, it burns a lot. (XX19, 14, M)*

*It is embarrassing to take your hands out of your pocket. It is also shameful to join other children when they eat food. When I was ill with scabies, I have never taken my hands out of my*

*pockets. When my friends invited me to eat food together, I would ask them to feed me with their own hands. I was afraid that they could be discomforted if I joined them because my hands were full of pus. I have been doing my best in order to make sure that no one sees my hands. (XX11, 15, M)*

Participants linked social consequences with psychological consequences:

*It is embarrassing to scratch your body in front of other people. Scratching in the presence of other people affects the impression that other people have about you (XX5, 14, M)*

*When you are infected with scabies, your skin swells and your friends seeing it will marginalize you because of fear of contamination. As a result, you will not only feel lonely, but also your self-confidence decreases. (XX2, 11, M)*

*I hate the moment other people laugh at me when I scratch my body. If you don't scroll (cover) your hand with a plastic bag, people will laugh at you and you become angry. (XX3, 14, M)*

Findings regarding the economic impact of scabies infestation were mixed. Some participants reported that scabies affected their economic wellbeing through restriction of movement and thus the amount of money they could obtain, and the opportunity cost of the money spent on medicine. On the other hand, others felt it could positively contribute to economic wellbeing as the illness and sick-role behavior could generate sympathy and more money from begging.

*It was after infection that I started to worry about the money needed to buy the medicine for it. When I realize that I have got enough money to buy the medicine, I then worry about how to pay for accommodation at night. (XX4, 17, F)*

*When infected, some street children do not want to recover from the disease: there are incidents of showing one's wounded hands to solicit more sympathy during begging. They beg, telling people that they needed the money for treatment and people give them without knowing the fact that they use it only for their addiction. (XX8, Male, 15)*

## Understandings about the epidemiology of scabies

Study participants believed that their susceptibility to scabies was linked to their living conditions. They believed that leaving the street would reduce susceptibility and many had experienced life without scabies infestation as a result of joining a childcare organization. The linkage the street children made between living conditions and infestation arose from their understanding about the causes of scabies:

*I had no alternative to sleeping together even if I knew that the child sleeping beside me was infected and could transmit scabies and other communicable diseases. (XX3, M, 12)*

*I don't believe that I would have been infected if I were at home. This is because, at home, I wear only my own clothes or the ones that belong to my sisters. (XX4, F, 17)*

*If I were at home, I would have been maintaining my personal hygiene and hence less likely to be exposed to scabies. (XX18, M, 15)*

*I was not infested with scabies for some time after entering an organization where I got a better chance of maintaining my personal hygiene. (XX5, M, 14)*

Further, they believed that exposure to dirty environments and lack of personal hygiene caused scabies. Accordingly, they attributed their susceptibility to being unable to maintain their personal hygiene:

*Scabies is related to the environment in which you live. For example, if you live in a dirty environment, you are likely to be infected with scabies. If your sleeping area has dust, it is more likely that you will be infected. (XX5, 14, M)*

*Initially, it is caused by dust. You will be infected when you are unable to wash your hand after touching the dust. If you don't wash your hand after eating food, you will soon start to itch. The same happens if you don't wash your hand after touching dust. (XX1, 16, M)*

*If you continue wearing the same clothes always, they breed lice. Then, the louse causes scabies (XX2, 11, M)*

*It has been over a month since I started wearing this trouser. I don't have the capacity to change my clothes. Therefore, I have neither changed my clothes nor taken a bath for a month. I have not washed my hair either and if I go through my hair, I can easily pick lice now. It is because of lack of water and people are also not willing to give us as they have to buy a jerrycan of water for 20 birr (0.36USD). (XX11, M, 15)*

Transmission of scabies was associated with the daily routines of living 'of the street' including repeated contact with others. Common street lifestyles, including sharing of personal materials such as clothes and addictive substances, were considered ways by which scabies was transmitted from one child to another:

*. . ...if I received and sniffed glue from a child who has been infected with scabies, it will be transmitted to me. (XX8, 15, M)*

It was after I borrowed someone's clothing that I started to see the symptoms of scabies (XX4, 17, F)

*It was transmitted to me through louse from a friend I have been sleeping with. (XX3, 12, M)*

Participants believed that scratching the areas of skin infested would cause further spread of the condition to the other parts of the body. They suggested that a person infected with scabies should not scratch to halt the spread of the disease:

*As you scratch it, that little swelling starts to secrete pus which distributes to other areas of the skin and reproduces more of its kind. (XX11, M, 15)*

*What you need to do is not to scratch it when you realize that you started to develop the inflammation. As you continue to scratch, it spreads to the other part of your body (XX5, 14, M)*

*A small wound appears on your skin and extends over to the other parts of the body with the fluid from the small wound as you scratch it. (XX1, 16, M)*

*One should not scratch at all. For example, I have an experience of drying some leftover wounds without scratching. Instead of scratching it, I used to take a cigarette from my smoker friends and burn the tip of the wound when it urges me to scratch. In the meantime, it dried without the need for further treatment. (XX11, M, 15)*

Street children's awareness about the disease and its causes is based on lived experience. Sharing information with those experiencing scabies for the first time was also described:

*I learned from my own experience of infection. When I was wearing my friend's clothes, other children were telling me not to wear them. Although their advice was based on the knowledge they got when infected with scabies, I ignored their advice and wore her clothes. It was after infestation that I became conscious about their appropriate advice. (XX9, F, 16)*

Although most participants had fair understanding of scabies, some showed misplaced knowledge, attitude and practice:

*When you write on your body using pen, the ink of the pen will be changed into scabies. Scabies is also caused by eating the food that you don't like. For example, if eating an egg is not convenient for your body, you will be infected with scabies when you eat an egg. (XX2, 11, M)*

*If another person laughs at the patient suffering from scabies, he will soon contract scabies. This implies that there is divine cause. In addition, it can be transmitted by being less considerate and not sympathetic to the person suffering from scabies. But, if you sympathize about the person's situation, you will be safe][this is akin to biblical phrase "don't judge, or you too will be judged"] (XX3, M, 14)*

*I contracted it from other children through breathing. (XX6, 14, M)*

## Health-seeking behavior

We uncovered both the primary (preventive) and secondary (illness response) health seeking behavior of the study participants in relation to scabies. Most informants believed that a person infected with scabies should seek treatment from modern healthcare facilities:

*As soon as you see the symptoms of scabies, you have to go to the health facilities where treatment can be given. (XX11, 15, M)*

*. . .he should visit a healthcare center and seek treatment. There is a medicine prepared specifically for this purpose and the physicians will prescribe for him. (XX8, 15, M)*

*. . ..he should visit healthcare facility and follow the prescription of the health professional. If he is unable to go to health center by himself, he should ask the support of other persons around him. (XX3, 12, M)*

The sick-role behavior of the children revealed common patterns of response to scabies infestation, predominantly by visiting modern healthcare facilities. Going to a health center or hospital where treatment was provided free of charge, or buying ointments from a nearby pharmacy have been identified as common responses:

*Because I was severely affected, I cried and asked my friends to support me. They took me to a hospital where the health workers removed my hair and I stayed for three days there. They then discharged me after 3 days giving me a white-colored ointment which helped me to cure the disease. (XX3, 12, M)*

*Whenever I am ill with any kind of disease, either I go to a health center or visit a nearby pharmacy and tell the pharmacists that I am suffering from this or that disease and ask them*

*to give me anti-pain. As soon as I receive the medicine, I will start using it in front of them. (XX11, M, 15)*

Study participants' experience of visiting modern healthcare facilities included identification of the type of medicine prescribed and understanding how the medicine is to be applied:

*The ointment should only be applied at nighttime or early morning: if you do it in the afternoon, it aggravates the disease. (XX3, 14, M)*

The experience of visiting a modern healthcare facility was found to be associated with the following circumstances:

## The role of mutual support or social capital as a street sub-culture

The study identified the presence of strong mutual support or social capital as a sub-culture of street life in the study area. In addition to sharing various resources, the sub-culture of street life involves taking care of one another in times of crisis, including health problems. Such mutual support mechanisms at the time of illness were found to promote health-seeking behavior:

*Here, there is a strong tradition of supporting each other when a child is affected with a certain health problem: some take the duty of fetching water for him while others engage in collecting money for transportation. The patient will then be taken to a health center for treatment. (XX11, 15, M)*

*With financial support from my friends, I went to a pharmacy where I bought an ointment for 150 birr. Based on the pharmacist's advice, I was gently applying the ointment on the infected areas and I recovered soon as a result. As soon as you are affected by any disease, children on the street take you to a health center. Like non street people in other areas, they don't leave you alone. This is a common experience among the street children. (XX9, 16, F)*

## The role of free healthcare services

Availability of free healthcare services was another factor that motivated street children to prioritize modern healthcare in their response to illness. Study participants disclosed that there were [public] hospitals in Addis Ababa where street children were entitled to free healthcare services for any health problem, including scabies. All they had to do was find the money to travel to the hospital to access healthcare. Medicine from a nearby pharmacy or other shop was also available without the need for diagnosis or prescription. Begging was reported to be the main source of funds to buy medicine in this way:

*As soon as you see the symptoms of scabies, you have to go to the health facilities where treatment can be given. There are two hospitals- Zewditu and Yekatit 12 (sister's clinic) where street children get free access to healthcare when affected with any kind of disease. (XX11, 15, M)*

*My friends told me that I would rather find some money to buy medicine from pharmacy than going to Sidist Kilo. Before you called me for the interview, I have been begging money at the traffic lights. If I get enough money, I will go to a pharmacy and buy the medicine. (XX4, 17, F)*

### Anticipated and experienced negative social reactions from significant others

The crack-like burrows appearing between the fingers seem to have made some study participants aware of the physical changes brought about by scabies. The psychosocial consequences of scabies infection, such as feelings of embarrassment and experiences of marginalization from valued social activities have been discussed above. Children of the street who aspire to return home reported to have worried about the physical changes and the possible reactions of family members, inspiring them to seek treatment:

> I thought that it would be embarrassing to return home in this condition having left home with a healthy body. It was after a few days of infection that I went to a health center for treatment. (XX8, Male, 15)

### Use of Alternative medicine

Some of the informants reported to have used the alternative treatments. Self-care or visiting traditional medical practitioners were the common alternative treatments for scabies:

> When I was infected with scabies, a woman told me to take a bath. The first time I heard from her, I didn't believe that it would work. But I repeatedly took a bath anyway to give it a try and I recovered. I saw that some street children wrongly bathed in river water. Because it can be polluted, it often aggravates their scabies. In my case, however, I took a bath with clean water which helped me recover. (XX5, 14, M)

> If you mix lemon and ginger and drop it on, it burns and finally the wound disappears. To get the intended outcome from this treatment, however, you have to first wash your body. Otherwise, the dirt of your body would affect the effectiveness of the treatment. (XX2, 11, M)

> Many will go to both sister's clinic and Mamo. The latter is a traditional healer of scabies who treats it at relatively low cost. I had an experience of using his traditional medicine that I found it effective. He gives you a liquid medicine in a plastic bag. I have been applying it on my skin at night and wash it in the morning. (XX4, 17, F)

### Preventive activities or behavior

Street children not only seek modern and alternative treatment services in response to scabies infestation, they also engage in activities to prevent the condition. Accordingly, keeping away from children infected with scabies, maintaining personal hygiene, keeping the hair short and avoiding exchanging clothes are considered to be common preventive health behaviors:

> One mechanism is wearing clean clothes. You have to also take a shower at least three to four times a week. In addition, you have to stay away from polluted environments. (XX5, 14, M)

> If one of the children gets infected with scabies, it is inevitable that he will spread it to the other children unless he remains cautious. Especially when you see that their hair grows and starts to contain lice, you have to advise them to go to a barber. There are many barber houses such as Mexico where free haircut services are given for us and there were occasions I took some of my friends there to get the service.(XX1, 16, M)

> . . .it has to do with maintaining one's personal hygiene which involves taking a bath and wearing clean clothes. (XX21, 13, M)

Study participants believed that their susceptibility to scabies was related to living conditions on the street and that they might have been less susceptible had they been at home. Some believed that reunification with their family or joining a childcare organization would help prevent scabies infestation, as these alternatives would give them better access to the resources needed to maintain personal hygiene:

> . . ..Street children like me should maintain their personal hygiene and return to their families if possible. They should ask their parents an apology and return home. For us, not only it is difficult to maintain our personal hygiene, but also we are not mostly interested to do so. It is, however much easier to maintain hygiene when living with one's family. Therefore, street children should ask their parents an apology and return home. (XX3, 12, M)

> If possible, the concerned bodies should try to return the children to their families. But if the children are not willing to rejoin their family, they should enable them to enter an organization where they have better access to the resources to maintain their personal hygiene and recover from addictions. (XX9, F, 16)

Participants' awareness about preventive mechanisms and their readiness to prevent the condition mainly came from their previous experience of infestation with the disease, including the experiential understanding of its epidemiology:

> It is for the first time that I took someone's clothes; I won't take clothes from other children again. This will be the last time for me to borrow clothes from other street girls. (XX4, 17, F)

### Challenges to act on preventive activities

Living on the street is linked to lack of resources which affects street children's efforts to prevent or respond to scabies. Study participants reported that shortage of money, peer pressure, discrimination by some healthcare professionals, shortage of clothes, and addiction to substances were common challenges affecting their health-seeking behavior:

> The lack of money and discrimination from the health professionals are the major barriers. I have an experience when taking my friends to a health center that some health professionals are afraid of contamination. In fact, there are also kind people who buy the medicine for us. (XX9, F, 16)

> When you try to take a bath, other children discourage you for taking the disease seriously. They want you to be like them when it comes to maintaining personal hygiene. I often wonder if I was really trying to be different from the other children around. (XX5, 14, M)

> Sniffing glue poses the greatest challenge to preventing scabies. After sniffing glue and becoming high, you will lose consciousness and feel tired. Sleeping wherever you can, it makes you not protect yourself from scabies. (XX11, 15, M)

### Discussion

This study showed that street children in Addis Ababa are highly susceptible to scabies. The children believe that they are more susceptible to scabies than children living with their family. They also reported frequent infestations. While our study was limited in some respects by the relatively small number of informants, specific settings potentially limiting generalizability,

and the likelihood of social desirability bias linked to the expectation of support, our main findings are important for these settings.

Although there are very few studies conducted on scabies among street children in Ethiopia, the findings of our study are consistent with studies undertaken in other countries. For instance, a study conducted in Kenya [34] found scabies to be the most common skin disease among children of the street. A comparative cross-sectional study conducted in Mumbai city [24] also found that children living on the street had significantly higher scabies morbidity than children living in NGO-run open houses. Other Ethiopian children living in difficult circumstances that put them at high risk of scabies infestation also reported that scabies was common among the study population. Typical examples could be studies undertaken among children attending religious education-"Yekolo Temaris" of the Orthodox Christian Church [26,35,36] and "Madresahs" of the Mosque [37] which concluded that scabies was a public health problem among these study groups.

Our study also indicated that street children linked their susceptibility to scabies with their living conditions. These include sleeping arrangements, which inevitably involve physical contact between the children (as they share bed sheets), lack of resources to maintain personal hygiene, sharing of personal belongings including clothes and glue, and addiction to substances. The study participants believed that while these situations increased the risk of transmission, lice were the main agents of transmission. This could be taken as another misconception held among the participants as they did not distinguish mites from lice. According to Lalor et al. [38], almost all the health problems reported by street children are related to their living arrangements, which included poor personal hygiene, overcrowding and inadequate water supply. Okoronkwo [39] also concluded that scabies infection was associated with overcrowding, infrequent bathing and scarcity of water. In addition, Karim et al. [40] found that poor sanitary conditions (including not washing clothes and infrequent bathing with soap), overcrowded sleeping arrangements, and sharing of clothes were associated with disease severity and re-infection among children living in densely populated areas of Bangladesh.

A community-based study in Ethiopia [13] found factors such as using only water (with no soap or detergents) for hand washing, having a contact history of scabies or skin lesions, and sharing beds were significantly associated with scabies. Another study conducted among primary school children by Hassen et al. [41] revealed that lower grade levels, children of uneducated fathers, and the lack of bathing with water and soap were significantly associated with scabies infestation. While the results of the present study in relation to the causes of scabies are consistent with most previous related studies, they differ from the findings of Ayaya and Esamai in Kenya [34]. The authors associated the high prevalence of scabies with inability to afford products such as Vaseline. They concluded that other groups of street children in their study (the African-Americans) were less susceptible to scabies because they had better access to petroleum products which can smother the scabies mite.

Street children in our research disclosed that the level of scabies infestation increased during the dry season and decreased during the rainy season. Though seasonal fluctuation in scabies infestation has also been found in previous studies, many of these studies (undertaken on different population groups) found that scabies morbidity increased during the cold weather. For instance, Karim et al. [40] noted that 96% of children were affected to a greater extent during cooler months. According to Armitage et al. [42], seasonal variability in skin infection may reflect behavioral changes between seasons or environmental changes with the possibility of increasing bacterial proliferation on skin. This explanation aligns with our study participants' perceptions that during the dry season, dust increases and so does contamination with it. In resource-limited settings such as Ethiopia, shortage of water increases during the dry season

and so street children's access to water decreases. This worsens already poor hygiene practices, enhancing their susceptibility to the scabies and other skin infections. On the other hand, other studies such as [42, 43] found high morbidity of skin infection, including scabies, during the rainy season which was explained by the fact that people spend more time in overcrowded indoor settings, thus increasing skin-to-skin contact. Korycińska et al. [43] also associated it with greater incidence of sexual activity during the cold season, which increases contact and the risk of scabies transmission. Nevertheless, further studies, especially with longitudinal design, are needed to explore seasonal variability in scabies morbidity among street children.

Visiting modern healthcare facilities was the most common health-seeking behavior, while using traditional medicine or self-care occurred less commonly. The financial and physical accessibility of modern healthcare facilities, the negative social reactions anticipated or actually experienced from relatives and the existence of mutual support were found to be factors promoting health-seeking behaviors. These findings are consistent with another study conducted in the same setting [38] which identified that the majority of street children (72.4%) go to medical centers, while 8.1% use traditional medicine, and 8% said they have nowhere to go. The Missionaries of Charity or "Sister-bayt" provided them with medical treatment, which was also reported in the present study. A study conducted in Dhaka city [44] found that the majority (68.1%) of street children received treatment from a medicine shop, 12.9% from a faith-healer, 33.6% through homeopathy, 11.2% from *kabiraj (traditional practitioner)*, 28.4% from hospital, very few (8.6%) from a qualified doctor, 3.4% private clinic and 4.3% from other sources. Our earlier study [25], demonstrated the efforts street children exert to maintain their health under constraining circumstances through personal hygiene and physical exercise.

These findings differ from those of Kassa [45] who concluded that street children in Addis Ababa did not utilize public healthcare facilities because of financial constraints, negative attitudes of healthcare professionals towards the street children, and children's perceptions of their health. While health professionals' negative attitudes were a barrier to access for some street children in our study, most had considerable experience of utilizing modern healthcare facilities, including public hospitals that provide free healthcare services. The difference could be attributed to chronological differences between the two studies in the sense that the former study was conducted before the enactment of many national policy directives aimed to improve access to social services for the urban destitute. Examples include the National Children's Policy (2017), the Health Care Service Guideline of Urban Destitute Project (2022), and the National Social Protection Policy (2012). Provision of free healthcare services in some public hospitals in the study area was one factor promoting the health-seeking behavior of street children. Further study is needed to find out whether free access to healthcare services for the street children is limited to Addis Ababa or applicable in other parts of the country too. This will necessitate identifying regional differences in the implementation of national guidelines, and the presence of local initiatives to enhance the wellbeing of street children.

## Conclusion

This study suggests that scabies is common in this neglected social group. Belief in modern healthcare facilities as a response to scabies and visiting both modern and traditional practitioners was reported. We recommend interventions that focus on supporting street children to leave the street and reintegrate with their family or the community of their choice. In this regard, both governmental and non-governmental organizations working on street children should strengthen interventions focusing on sustainably reunifying and reintegrating street children. We hope that the findings will contribute towards efforts to fight health inequality and neglect.

## Author Contributions

**Conceptualization:** Bewunetu Zewude, Getnet Tadele, Gail Davey.

**Data curation:** Bewunetu Zewude, Getnet Tadele, Gail Davey.

**Formal analysis:** Bewunetu Zewude, Getnet Tadele.

**Funding acquisition:** Getnet Tadele, Gail Davey.

**Investigation:** Bewunetu Zewude, Getnet Tadele, Gail Davey.

**Methodology:** Bewunetu Zewude, Getnet Tadele.

**Project administration:** Getnet Tadele, Gail Davey.

**Resources:** Getnet Tadele, Gail Davey.

**Software:** Bewunetu Zewude.

**Supervision:** Bewunetu Zewude, Getnet Tadele, Gail Davey.

**Validation:** Bewunetu Zewude, Getnet Tadele, Gail Davey.

**Visualization:** Gail Davey.

**Writing – original draft:** Bewunetu Zewude.

**Writing – review & editing:** Bewunetu Zewude, Getnet Tadele, Gail Davey.

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
