## [Decision Letter · Decision Letter 0]

28 May 2024

Dear Mr. Zewude,

Thank you very much for submitting your manuscript "Neglected Tropical Disease meets Neglected Community: Street Children’s Susceptibility to Scabies in Addis Ababa, Ethiopia" for consideration at PLOS Neglected Tropical Diseases. As with all papers reviewed by the journal, your manuscript was reviewed by members of the editorial board and by several independent reviewers. In light of the reviews (below this email), we would like to invite the resubmission of a significantly-revised version that takes into account the reviewers' comments. 

We cannot make any decision about publication until we have seen the revised manuscript and your response to the reviewers' comments. Your revised manuscript is also likely to be sent to reviewers for further evaluation.

Sincerely,

Aysegul Taylan Ozkan, M.D., Ph.D.,

Academic Editor

Audrey Lenhart

Section Editor

Reviewer's Responses to Questions

**Key Review Criteria Required for Acceptance?**

**Methods**

-Are the objectives of the study clearly articulated with a clear testable hypothesis stated?

-Is the study design appropriate to address the stated objectives?

-Is the population clearly described and appropriate for the hypothesis being tested?

-Is the sample size sufficient to ensure adequate power to address the hypothesis being tested?

-Were correct statistical analysis used to support conclusions?

-Are there concerns about ethical or regulatory requirements being met?

Reviewer #1: Selection of study participants 

1. The authors talk of collecting data about the embodied experience of scabies infestation, street children who had a history of a previous scabies infestation or those who were infested with it during the interview. Was scabies diagnosed clinically (clinician) or just the mere presence of scabies-like skin manifestation? The reader needs to be sure that these street children actually had scabies and not the differential diagnosis (other skin infestations) of scabies.

2. It will be useful for the authors to list the inclusion/exclusion criteria first before explaining each of them.

Method of data collection and analysis

1. The statement “Qualitative data were collected using an in-depth interview method” should read “Qualitative data was collected using in-depth interviews”

2. It will be important to indicate what is contained in the interview guide or refer the reader to it as a supplementary file.

3. Was the interview guide written in the local language or in another language and translated? If you so was the translation done by a professional or how was it done?

4. Were the interviews conducted in the entire December 2023? The authors need to be specific with the dates.

5. The statement “The number of children interviewed was determined based on the principle of theoretical data saturation” is not clear. The need to explain what this theory means in the context of this article and indicate the source as well.

6. Why were most participants getting bored after some minutes of conversation?

7. The data analysis section needs to be rewritten well. For instance how did the authors manage issues of disagreement during coding? It is also not clear how the main themes and sub-themes were arrived at.

8. Was data analysis done inductively or deductively? This needs to come out clearly.

Reviewer #2: (No Response)

Reviewer #3: Methods

The study was conducted in Addis Ababa, focusing on Bole and Kirkos sub-cities. Participants were selected using snowball sampling, with inclusion criteria ensuring diverse representation of street children aged 11-17. The qualitative data were collected through in-depth interviews, and the analysis involved coding and thematic analysis using NVivo 12 software. Ethical considerations were carefully addressed, given the vulnerability of the study population.

Reviewer #4: The study design has been appropriate and the study population has been described. 

sample size is a limitation. Nevertheless, the authors have tried to summarize the views of the partcipants

Concerns about ethical and regulatory requirements met

**Results**

-Does the analysis presented match the analysis plan?

-Are the results clearly and completely presented?

-Are the figures (Tables, Images) of sufficient quality for clarity?

Reviewer #1: 1. It will be important to include some percentages in the socio-demographic characteristics.

2. It will also be useful for the authors to organize the results into main and sub-themes and clearly stated as such.

3. This section is poorly organized. 

4. It is better to present some of the results as figures or percentages so that the reader gets to know what proportion of the participants expressed their views of a particular finding.

Susceptibility to Scabies

1. Please check “•••”(‘ekek’).

2. What makes it clear that street children in Addis Ababa are susceptible to scabies?

3. The authors use the term prevalence loosely. How is susceptibility related to prevalence? The authors must realize that prevalence is a technical term and should be treated as such.

4. Not too sure how physical, social and psychological impacts fall under susceptibility. Similarly same applies to social consequences with psychological consequences.

5. Why talk about prevalence here when the objective of the paper was to explore the susceptibility to scabies and related health-seeking behavior of street children in Addis Ababa?

Did the headings: The role of mutual support or social capital as a street sub-culture; The role of free healthcare services; and Anticipated and experienced negative social reactions from significant others emerge from the data or were predetermined?

Discussion

1. The statement “This study showed that street children in Addis Ababa and possibly in the entire country are highly susceptible to scabies” cannot be arrived at from this type of study.

2. The statement “They reported frequent infestations, which was borne out by observation” is not clear.

3. Since this is not a prevalence study it is inappropriate to compare prevalence of scabies in other areas to that of this study.

Reviewer #2: (No Response)

Reviewer #3: The results section effectively presents the findings, categorized into themes such as susceptibility to scabies, understanding of its causes and symptoms, and health-seeking behaviors. The narrative is enriched with quotes from the participants, providing a vivid account of their lived experiences. The study reveals that street children are highly susceptible to scabies, with significant physical, psychological, and social impacts. The children’s understanding of scabies is linked to their living conditions and hygiene practices, and their health-seeking behavior includes visiting healthcare facilities, traditional healers, and self-care.

Reviewer #4: In the results paragraph, details pertaining to the total number, number and percentage of boys vs girls, youngest age, oldest age have to be mentioned. 

results have been presented in a narrative pattern

**Conclusions**

-Are the conclusions supported by the data presented?

-Are the limitations of analysis clearly described?

-Do the authors discuss how these data can be helpful to advance our understanding of the topic under study?

-Is public health relevance addressed?

Reviewer #1: 1. The study design and limitations of this study does not allow for the statement that “This study has uncovered considerable morbidity ascribable to a neglected tropical disease among a neglected social group”

2. The conclusion should be a brief summary of key findings.

3. The recommendation is not specific and non-targeted. Whom are the recommendations meant for?

Reviewer #2: (No Response)

Reviewer #3: Conclusion

The conclusion highlights the significant morbidity caused by scabies among street children and underscores the need for targeted interventions to improve their living conditions. The study calls for efforts to reunify street children with their families or integrate them into supportive communities, aiming to reduce health disparities and combat neglect.

Reviewer #4: The authors have highlighted the immediate need to address this NTD infection with due importance given to rejoining the street children to their family/homes and deaddiction from glue.

There is an urgent need to find out if the street children in the other sates have access to treatment care as per National policy guidelines.

The results indirectly point out to the need to address water scarcity.

**Editorial and Data Presentation Modifications?**

Reviewer #1: (No Response)

Reviewer #2: (No Response)

Reviewer #3: Minor revision

Reviewer #4: The authors have taken up this study with a good intention. I have mentioned a few corrections alongside the manuscript, which is in a narrative format. 

The manuscript could be made more concise . References need to be inserted, rather than mentioning the authors within the brackets

**Summary and General Comments**

Reviewer #1: 1. The manuscript is saddled with lots grammatical errors and will need substantial English editing.

2. The referencing style of the manuscript doesn’t appear to be that recommended by Plos journals. 

3. Line numbering is recommended would have aided easy review.

4. What is the quote (just before background) of page 2 meant for? 

5. The entire paper needs extensive formatting.

Background 

1. It will be useful to indicate the current prevalence of scabies in Ethiopia instead of stating scabies as being the most common diagnosis in the statement “weighted prevalence of clinically confirmed skin diseases in Ethiopia is 22.5%, with scabies being the most common diagnosis”

2. In the last paragraph the authors state that “The few existing studies are dominated by medical perspectives, and studies undertaken from social science perspectives are lacking”. This justification is not clear or convincing enough. Social science perspective is too broad and hence the need for the authors to be very clear on this.

Reviewer #2: (No Response)

Reviewer #3: General Comments

Relevance: The study addresses a critical intersection of neglected diseases and marginalized populations, providing valuable insights for public health policy and intervention strategies.

Clarity and Structure: The article is well-organized and clearly written, making it accessible to both researchers and policymakers

Recommendations: The recommendations are practical and actionable, focusing on improving living conditions and healthcare access for street children.

Ethical Considerations: The study demonstrates a strong ethical framework, ensuring the safety and confidentiality of the vulnerable participants.

Overall, the article is a significant contribution to understanding and addressing the health challenges faced by street children in Addis Ababa, particularly in relation to scabies.

Reviewer #4: Please address the comments mentioned alongside the manuscript. The manuscript could be made more concise . References need to be inserted, rather than mentioning the authors within the brackets

References could be mentioned in Vancouver style

PLOS authors have the option to publish the peer review history of their article (what does this mean?). If published, this will include your full peer review and any attached files.

Reviewer #1: Yes: Stephen Apanga

Reviewer #2: Yes: Aslan Yürekli

Reviewer #3: Yes: Dr.Fantahun Abza Babeta, Assistant professor, MD, Dermatologist

Reviewer #4: No
---

## [Decision Letter · Decision Letter 1]

21 Jul 2024

Dear Mr. Zewude,

Thank you very much for submitting your manuscript "Neglected Tropical Disease meets Neglected Community: Street Children’s Susceptibility to Scabies in Addis Ababa, Ethiopia" for consideration at PLOS Neglected Tropical Diseases. As with all papers reviewed by the journal, your manuscript was reviewed by members of the editorial board and by several independent reviewers. The reviewers appreciated the attention to an important topic. Based on the reviews, we are likely to accept this manuscript for publication, providing that you modify the manuscript according to the review recommendations. 

Sincerely,

Aysegul Taylan Ozkan, M.D., Ph.D.,

Academic Editor

Audrey Lenhart

Section Editor

Reviewer's Responses to Questions

**Key Review Criteria Required for Acceptance?**

**Methods**

-Are the objectives of the study clearly articulated with a clear testable hypothesis stated?

-Is the study design appropriate to address the stated objectives?

-Is the population clearly described and appropriate for the hypothesis being tested?

-Is the sample size sufficient to ensure adequate power to address the hypothesis being tested?

-Were correct statistical analysis used to support conclusions?

-Are there concerns about ethical or regulatory requirements being met?

Reviewer #1: (No Response)

Reviewer #3: Yes

**Results**

-Does the analysis presented match the analysis plan?

-Are the results clearly and completely presented?

-Are the figures (Tables, Images) of sufficient quality for clarity?

Reviewer #1: (No Response)

Reviewer #3: Yes

**Conclusions**

-Are the conclusions supported by the data presented?

-Are the limitations of analysis clearly described?

-Do the authors discuss how these data can be helpful to advance our understanding of the topic under study?

-Is public health relevance addressed?

Reviewer #1: (No Response)

Reviewer #3: Yes

**Editorial and Data Presentation Modifications?**

Reviewer #1: (No Response)

Reviewer #3: Yes

**Summary and General Comments**

Reviewer #1: (No Response)

Reviewer #3: No comments

PLOS authors have the option to publish the peer review history of their article (what does this mean?). If published, this will include your full peer review and any attached files.

Reviewer #1: Yes: Stephen Apanga

Reviewer #3: Yes: Dr.Fantahun Abza Babeta, MD Dermatologist

Figure Files:

Data Requirements:

Reproducibility:

References

---

## [Decision Letter · Decision Letter 2]

19 Aug 2024

Dear Mr. Zewude,

We are pleased to inform you that your manuscript 'Neglected Tropical Disease meets Neglected Community: Street Children’s Susceptibility to Scabies in Addis Ababa, Ethiopia' has been provisionally accepted for publication in PLOS Neglected Tropical Diseases.

Best regards,

Aysegul Taylan Ozkan, M.D., Ph.D.,

Academic Editor

Audrey Lenhart

Section Editor

Reviewer's Responses to Questions

**Key Review Criteria Required for Acceptance?**

**Methods**

-Are the objectives of the study clearly articulated with a clear testable hypothesis stated?

-Is the study design appropriate to address the stated objectives?

-Is the population clearly described and appropriate for the hypothesis being tested?

-Is the sample size sufficient to ensure adequate power to address the hypothesis being tested?

-Were correct statistical analysis used to support conclusions?

-Are there concerns about ethical or regulatory requirements being met?

Reviewer #1: (No Response)

**Results**

-Does the analysis presented match the analysis plan?

-Are the results clearly and completely presented?

-Are the figures (Tables, Images) of sufficient quality for clarity?

Reviewer #1: (No Response)

**Conclusions**

-Are the conclusions supported by the data presented?

-Are the limitations of analysis clearly described?

-Do the authors discuss how these data can be helpful to advance our understanding of the topic under study?

-Is public health relevance addressed?

Reviewer #1: (No Response)

**Editorial and Data Presentation Modifications?**

Reviewer #1: (No Response)

**Summary and General Comments**

Reviewer #1: (No Response)

PLOS authors have the option to publish the peer review history of their article (what does this mean?). If published, this will include your full peer review and any attached files.

Reviewer #1: **Yes: **Stephen Apanga

---

## [Editor Report · Acceptance letter]

2 Sep 2024

Dear Mr. Zewude,

We are delighted to inform you that your manuscript, "Neglected Tropical Disease meets Neglected Community: Street Children’s Susceptibility to Scabies in Addis Ababa, Ethiopia," has been formally accepted for publication in PLOS Neglected Tropical Diseases.

Best regards,

Shaden Kamhawi

co-Editor-in-Chief

Paul Brindley

co-Editor-in-Chief
